# Winter Malting Barley Growth, Yield, and Quality following Leguminous Cover Crops in the Northeast United States

Arthur Siller [1,*], Heather Darby [2], Alexandra Smychkovich [1] and Masoud Hashemi [1,*]

1   Stockbridge School of Agriculture, University of Massachusetts, Amherst, MA 01003, USA; asmychko@umass.edu
2   Department of Plant and Soil Science, University of Vermont, St. Albans, VT 05478, USA; Heather.Darby@uvm.edu
*   Correspondence: asiller@umass.edu (A.S.); masoud@umass.edu (M.H.)

**Abstract:** There is growing interest in malting barley (*Hordeum vulgare* L.) production in the Northeastern United States. This crop must meet high quality standards for malting but can command a high price if these quality thresholds are met. A two-year field experiment was conducted from 2015 to 2017 to evaluate the impact of two leguminous cover crops, sunn hemp (*Crotalaria juncea* L.) and crimson clover (*Trifolium incarnatum* L.), on subsequent winter malting barley production. Four cover crop treatments—sunn hemp (SH), crimson clover (CC), sunn hemp and crimson clover mixture (SH + CC), and no cover crop (NC)—were grown before planting barley at three seeding rates (300, 350, and 400 seeds m$^{-2}$). SH and SH + CC produced significantly more biomass and residual nitrogen than the CC and NC treatments. Higher barley seeding rates led to higher seedling density and winter survival. However, the subsequent spring and summer barley growth metrics, yield, and malting quality were not different in any of the treatments. There is much left to investigate in determining the best malting barley production practices in the Northeastern United States, but these results show that winter malting barley can be successfully integrated into crop rotations with leguminous plants without negative impacts on barley growth, yield, and grain quality.

**Keywords:** winter malting barley; malting quality indices; summer cover crops; sunn hemp; crimson clover; seeding rate; nitrogen management

## 1. Introduction

Winter malting barley (*Hordeum vulgare* L.) is an emerging crop in the Northeastern United States [1,2]. Although it has the potential to be a profitable crop in the region, achieving the high-quality grains needed for malting purposes is challenging because of the Northeast's humid environment and seasonal temperature extremes [3]. However, if the barley meets the malting quality standards, there is a price premium compared with feed grain [4] and the potential for additional local markets in the regional malting and brewing industry [5,6].

A successful malting barley crop must grow well, produce good yields of high-quality grain, and be harvested and stored correctly to maintain quality [7]. To be acceptable for malting, the barley grains should be large, low in protein, free of or very low in carcinogenic deoxynivalenol (DON) toxin [8], and sprout well during the malting process [9]. Farmers can successfully grow malting barley by combining three methods: (1) choosing a site-appropriate variety that will overwinter, resist locally common diseases, and remain upright after heading [10,11]; (2) correctly timing their harvest to avoid partial sprouting in the field and using forced air dryers if weather does not permit dry-down in the field [2,12]; and (3) using growing practices that have been shown to promote good malting quality [1,13].

Grain size is quantified by three metrics: test weight, percent plump, and percent thin. High test weight and high percent plump indicate that the kernels are large and

relatively uniform, while high percent thin means that the kernels are small with little energy for the malting process [7]. For the grains to sprout well during malting, they should have greater than 95% germinative energy and they must have a falling number greater than 250 s, indicating that they have not pre-sprouted in the field or during storage. Deoxynivalenol (DON) content must be below 0.5 mg kg$^{-1}$ while protein should be below 125 g kg$^{-1}$ [10,12]. While regional weather variability may prevent farmers from achieving malt-quality harvests in every year [2], the growing body of research into winter malting barley production can minimize this production risk for growers.

Previous research has produced varietal recommendations for malting barley growing in the Northeast [10,11] as well as recommendations for planting dates and fertilization rates [13,14]. However, winter malting barley is still a relatively new and minor crop in the Northeast [15] and most scientific reports are related to the drier areas of western North America.

In addition to selecting appropriate varieties and awareness of agronomic recommendations, it is essential that farmers understand how malting barley interacts with other crops in their rotations [16] and how other growing practices could influence crop rotation decisions. Since nitrogen can affect many aspects of barley malting quality, nitrogen cycling is of particular interest when considering how to integrate malting barley into a larger crop rotation. High levels of nitrogen fertilization can increase malting barley yields and grain size [17,18] but can also lead to excessive protein content [13,17–20], lower nitrogen use efficiency [13,17,19,21], and lower falling number [13]. While leguminous crops can contribute substantial amounts of nitrogen to subsequent crops, it is unknown whether nitrogen from legumes would have the same effects on winter malting barley as soluble nitrogen sources.

Since barley has the potential for producing a large number of tillers in the spring, the final yield does not respond linearly to increased planting density [22,23]. However, higher seeding rates may counteract the effect of excess nitrogen since higher seeding rates can reduce protein concentration and grain size [20,24]. The impact of crop rotation patterns may also be more noticeable in the fall when the plants are small [23] and different seeding rates may be differentially productive following a nitrogen-producing legume than a summer fallow.

Sunn hemp (*Crotalaria juncea* L.) and crimson clover (*Trifolium incarnatum* L.) are grown in the Northeast as summer forages or cover crops and can fit well into short growing periods before winter barley planting in the fall [25,26]. Farther north in eastern North America, Darby et al. [23] reported that barley yield was lower following sunn hemp but its malting quality was not affected, while crimson clover did not impact barley growth relative to summer fallow. In western North America, winter malting barley has also performed well following peas (*Pisum sativum* L.) and canola (*Brassica napus* L.) [16,17] but these crops are not commonly grown in the Northeast, and local rotation recommendations are needed. Whether grown as forages or cover crops, sunn hemp and crimson clover can have many impacts on agricultural productivity and ecosystem services. They can protect the soil from erosion [27], contribute organic matter to the soil [26], reduce insect pest damage [28,29], provide income if harvested as a forage [26,27], and add plant-available nitrogen to the soil [27,30].

In many rotations, the nitrogen contribution from sunn hemp or crimson clover would be beneficial to the following crop, but this may not be the case for winter malting barley if nitrogen from the preceding crop leads to excessive grain protein or otherwise reduces malting quality. Alternatively, if nitrogen from leguminous cover crops does not negatively affect malting barley, this would suggest that farmers can plant winter malting barley with minimal concern about excess nitrogen contributions from preceding crops.

The current experiment examines how integrating legumes into winter malting barley cultivation can affect grain yield and quality, and whether the barley seeding rate changes these effects. This knowledge will complement previous varietal assessment and agronomic

management recommendations to help farmers improve their profitability and integrate winter malting barley into their overall farm systems.

## 2. Materials and Methods

Experimental Site: A two-year field experiment was performed at the University of Massachusetts Agricultural Experiment Station Farm in South Deerfield, MA (42° N, 73° W) on fine Hadley loam soil. In both years, the experimental crops were grown after summer corn silage (*Zea Mays* L.) and winter fallow. The top 15 cm of soil was analyzed before cover cropping each year and the fields were amended with lime, sulfur, and potassium as recommended by the University of Massachusetts Soil and Plant Nutrient Testing Laboratory (Amherst, MA) for barley production. The experiment site was prepared using disk tillage immediately before the first planting date. Extreme weather events did not appear to influence the experiment in either year (Table 1).

**Table 1.** Weather Data for the experimental site in 2015 and 2016.

| Year | Month | Avg Temp (°C) | Departure from Avg. * | Max Temp (°C) | Departure from Avg. | Min Temp (°C) | Departure from Avg. | Total Precipitation (cm) | Departure from Avg. |
|------|-------|------|------|------|------|------|------|------|------|
| 2015 | July | 21.1 | −1.3 | 32.7 | 4.0 | 11.3 | −4.8 | 8.4 | −2.1 |
| | August | 21.1 | −0.3 | 32.5 | 4.6 | 11.3 | −3.7 | 6.4 | −3.9 |
| | September | 18.3 | 0.9 | 33.0 | 8.8 | 4.9 | −5.7 | 16.3 | 4.9 |
| | October | 9.2 | −1.4 | 23.3 | 6.3 | −7.4 | −11.8 | 5.6 | −7.4 |
| | November | 6.2 | 1.7 | 23.1 | 12.7 | −8.9 | −7.5 | 5.1 | −3 |
| | December | 4.0 | 4.8 | 16.4 | 12.2 | −5.5 | 0.4 | 11.9 | 1.3 |
| 2016 | January | −2.7 | 1.6 | 11.0 | 9.7 | −15.5 | −5.5 | 3.8 | −3.7 |
| | February | −1.9 | 1.3 | 14.9 | 12.0 | −26.1 | −16.8 | 10.4 | 2.4 |
| | March | 4.7 | 3.2 | 25.5 | 18.0 | −8 | −3.5 | 8.4 | 0.1 |
| | April | 7.4 | −0.8 | 26.2 | 11.2 | −11 | −12.5 | 5.3 | −4.5 |
| | May | 14.2 | −0.2 | 32.6 | 11.6 | −1.7 | −9.5 | 6.6 | −2.7 |
| | June | 19.1 | −0.2 | 30.9 | 5.5 | 5.3 | −7.7 | 3.6 | −8.7 |
| | July | 22.3 | −0.1 | 34.4 | 5.7 | 9.9 | −6.1 | 4.3 | −6.2 |
| | August | 22.2 | 0.8 | 33.6 | 5.7 | 8.9 | −6.1 | 4.6 | −5.6 |
| | September | 17.7 | 0.3 | 30.6 | 6.4 | 2.7 | −8 | 9.3 | −2 |
| | October | 10.3 | −0.4 | 23.8 | 6.9 | −4 | −8.4 | 5.4 | −7.6 |
| | November | 4.2 | −0.3 | 19.0 | 8.6 | −6.5 | −5.1 | 8.2 | 0.1 |
| | December | −1.6 | −0.7 | 11.4 | 7.2 | −19.2 | −13.3 | 7.7 | −2.9 |
| 2017 | January | −1.3 | 3.0 | 13.6 | 12.2 | −18.7 | −8.7 | 7.0 | −0.5 |
| | February | −0.3 | 2.8 | 20.8 | 17.8 | −18.6 | −9.2 | 3.8 | −4.2 |
| | March | −0.8 | −2.4 | 15.4 | 7.9 | −14.4 | −10 | 4.0 | −4.3 |
| | April | 10.2 | 2.0 | 29.0 | 14.0 | −3.7 | −5.2 | 11.1 | 1.3 |
| | May | 13.0 | −1.4 | 33.3 | 12.3 | 0.2 | −7.6 | 8.2 | −1.1 |
| | June | 18.8 | −0.4 | 34.4 | 9.0 | 4.7 | −8.3 | 11.8 | −0.5 |
| | July | 20.6 | −1.8 | 32.3 | 3.6 | 10.7 | −5.4 | 5.7 | −4.8 |

\* Average weather data from Amherst, MA—eight miles from South Deerfield. Averages are based on the years 2001–2020.

Experimental Layout: Four replications of each treatment were planted in a randomized complete block design, with cover crop species and barley seeding rate as fixed main effects. The 12 treatments consisted of balanced combinations of three barley seeding rates (300, 350, and 400 seeds m$^{-2}$) and four summer cover crop species, including sunn hemp (SH), crimson clover (CC), sunn hemp and crimson clover (SH + CC), and summer fallow with no cover crop (NC).

Field Management and Assessments: Summer cover crops were planted on 19 July 2015 and 11 August 2016. The cover crops were planted at the following rates. SH: 33.6 kg ha$^{-1}$ sunn hemp, CC: 20.2 kg ha$^{-1}$ crimson clover, SH + CC: 16.8 kg ha$^{-1}$ sunn hemp and 16.8 kg ha$^{-1}$ crimson clover. Cover crop aboveground biomass was sampled from two 0.5 m$^2$ sections on 8 September 2015 and 15 September 2016. Cover crop nitrogen content was calculated from crude protein using near-infrared spectroscopy (NIR) (Inframatic 8600, Perten Instruments). Cover crop biomass analysis included weeds growing with cover crops. Cover crops were flail mowed and terminated using a rototiller on 15 September 2015 and 16 September 2016.

Wintmalt, a 2-row malting barley, was planted on 25 September 2015 and 30 September 2016. Wintmalt is the common winter barley grown in New England. Cover crops and barley seeds were planted two cm deep, using a custom-made plot-size cone seed drill with 17.8 cm between rows.

Barley stands were counted on 16 October 2015 and October 2016. However, stand count data from 2016 was lost and the reported barley stand counts are based solely on data from 2015. Fall soil nitrate was measured immediately following the first hard frost on 20 October 2015 and 17 November 2016. Winter survival was not assessed in the spring of 2016 and was measured on 28 April 2017 using a 0–10 scale with 10 as complete survival and 0 as totally winter killed. Soil samples were collected to assess spring soil nitrate using five 6-inch-deep cores per plot, air dried, and soil nitrate content was determined using a LaChat QuickChem 8500 Series 2 Flow Injection Analysis System [31]. An amount of 28 kg ha$^{-1}$ nitrogen was applied as calcium ammonium nitrate on 15 April in both 2016 and 2017.

Foliar disease was estimated on 10 July 2017 as a percentage of leaf surface area infected using the disease guides in the American Phytopathological Society's ''A Manual of Assessment Keys for Plant Disease'' [32]. Due to rapid drought-induced foliar desiccation, foliar diseases were not measured in 2016. Heading date was declared when half of the tillers had emerged heads and is reported as Julian date. Plant height was measured on 24 June 2016 and 10 July 2017 while lodging was assessed on 12 July 2016 and 10 July 2017. Lodging was visually evaluated on a 0–10 scale with 0 as no lodging and 10 as completely lodged.

Harvest and Laboratory Analyses: Barley was harvested on 19 July 2016 and 17 July 2017 using an ALMACO SPC20 plot combine. A subsample of the grain was dried in a forced air oven at 38 °C to preserve kernel integrity. Germinative energy, test weight, and 1000-kernel weight were determined using ASBC methods Barley–3A, Barley–2B, and Barley–2 [9]. 2017 grain samples could not be analyzed immediately following grain harvest and the samples had to be stored in a walk-in cooler for several years before analysis. As a result, germinative energy was considerably lower for these samples and, although their inclusion would not change the statistical results, they are not included in our results and analysis. Malting quality was assessed at the E.E. Cummings Crop Testing Laboratory at the University of Vermont (Burlington, VT). Crude protein content as a proportion of dry matter was measured with a Perten Inframatic 8600 Flour Analyzer, and falling number was assessed using the AACC Method 56–81B [33] on a Perten FN 1500 Falling Number Machine. DON content was evaluated in the subsamples using the NEOGEN Corp. Veratox DIN 2/3 Quantitative Test with a limit of detection of 0.1 mg kg$^{-1}$.

Statistical Analysis: Data were analyzed using the permlmer and lmer functions in the predictmeans [34] and lme4 [35] packages of R statistical software [36]. Permutation tests were used to assess the impact of the main fixed effects of *cover crop type* and *barley seeding rate* as well as their interaction at a significance level of $p \leq 0.05$. This non-parametric method was used to account for non-normal distribution of residuals and heterogeneous variance in many response variables. Bonferroni adjusted *t*-tests were used to make pairwise comparisons between all *cover crop type* treatments and orthogonal polynomial regression was used to assess the continuous effect of *barley seeding rate*. Because variance between groups was homogeneous, pooled standard deviations were used to calculate pairwise *t*-tests for *fall soil nitrate* and *barley grain falling number* while non-pooled standard deviations were used in the analysis of *cover crop biomass* and *cover crop nitrogen content* to account for heterogeneous variance. The random variables of *year* and *block* were combined into one random variable and data from both years were combined and analyzed collectively as eight replications.

## 3. Results

### 3.1. Cover Crop Biomass and Nitrogen Content

Although nitrogen concentrations in aerial parts of the cover crops were not significantly different, due to higher biomass production sunn hemp (SH) and sunn hemp-crimson clover mixture (SH + CC) produced significantly more biomass than either monocrop crimson clover (CC) or weedy fallow (NC) (Table 2, Figure 1). Therefore, their aboveground biomass residues added more nitrogen to the soil when incorporated before barley planting (Table 2, Figure 2). SH produced the highest amount of aboveground dry matter (2.90 t ha$^{-1}$), thus contributing the highest nitrogen (78.8 kg ha$^{-1}$). Both local weeds and CC treatments produced considerably lower dry matter (0.87 and 0.59 t ha$^{-1}$, respectively); therefore, their nitrogen contributions to the cropping system were minimal compared with SH and cover crop mixture (Figure 3). Interestingly, the local weeds in NC plots contained significantly more nitrogen (37.3 g kg$^{-1}$) than in monoculture and mix legumes (Figure 2). The results indicate that the local weeds were more aggressive and efficient in taking up nitrogen from the soil. The effective nutrient uptake by weeds from the soil likely demonstrates their capacity to adapt to changes in edaphoclimatic conditions during the growing season [37,38]. To our surprise, CC performed poorly in terms of atmospheric nitrogen fixation and accumulation in plants' aerial tissues. The failure of CC to fix nitrogen could be due to the poor rhizobium activity and thus requires further investigation.

**Table 2.** Mean cover crop dry matter biomass and nitrogen content following different cover crops in South Deerfield MA in 2014–2015 and 2015–2016.

| Cover Crop Type | Cover Crop Dry Matter Yield (t ha$^{-1}$) | | Cover Crop Nitrogen Content (g kg$^{-1}$) | | Cover Crop Nitrogen Yield (kg ha$^{-1}$) | |
|---|---|---|---|---|---|---|
| Sunn Hemp | 2.90 | a | 27.1 | b | 78.7 | a |
| Crimson Clover | 0.59 | b | 29.4 | b | 17.5 | c |
| Sunn Hemp and Crimson Clover | 2.24 | a | 25.6 | b | 57.4 | a |
| No Cover Crop | 0.87 | b | 37.3 | a | 32.5 | b |
| Overall Experiment Mean | 1.82 | | 27.3 | | 49.6 | |
| Effect Significance | | | | | | |
| Cover Crop Type | *** | | * | | *** | |

Note. *, $p \leq 0.05$; ***, $p \leq 0.001$; ns, non-significant according to non-parametric permutation tests. For significant effects, all pairwise comparisons were made using Bonferroni adjusted *t*-tests. Means followed by the same letter are not significantly different from each other ($p \leq 0.05$). Cover crop dry matter and nitrogen content includes weeds.

### 3.2. Soil Nitrate in the Fall and Spring

In the late fall, soil nitrate levels were nearly identical in CC (2.0 mg kg$^{-1}$) and NC (2.0 mg kg$^{-1}$) plots (Table 3). Soil nitrate was significantly higher in the SH plots (3.3 mg kg$^{-1}$), while SH + CC plots had intermediate soil nitrate (2.8 mg kg$^{-1}$) (Table 3). As expected, spring soil nitrate was substantially higher overall compared to fall soil nitrate and ranged from 5.9 to 7.2 mg kg$^{-1}$, depending on cover crop treatments. However, soil nitrate in spring was highly variable and the differences between treatments were not significant (Table 3).

### 3.3. Barley Seeding Rate Impact on Stand Establishment, Winter Survival, and Growth of Barley

Barley seedling populations increased with increased seeding rate (Table 4). There was a significant quadratic regression describing this relationship (Seedling Population = $-0.0105x^2 + 8.1424x - 1211$), showing that there was a leveling off of barley population between 350 and 400 seeds m$^{-1}$. This suggests that higher seeding rates do not increase barley populations towards the higher end of this range (Figure 4). Higher seeding rates also

significantly improved seedlings' winter survival (Winter Survival = 0.0313x + 86.25) but the difference was small and all seeding rate levels had better than 95 percent winter survival (Figure 5).

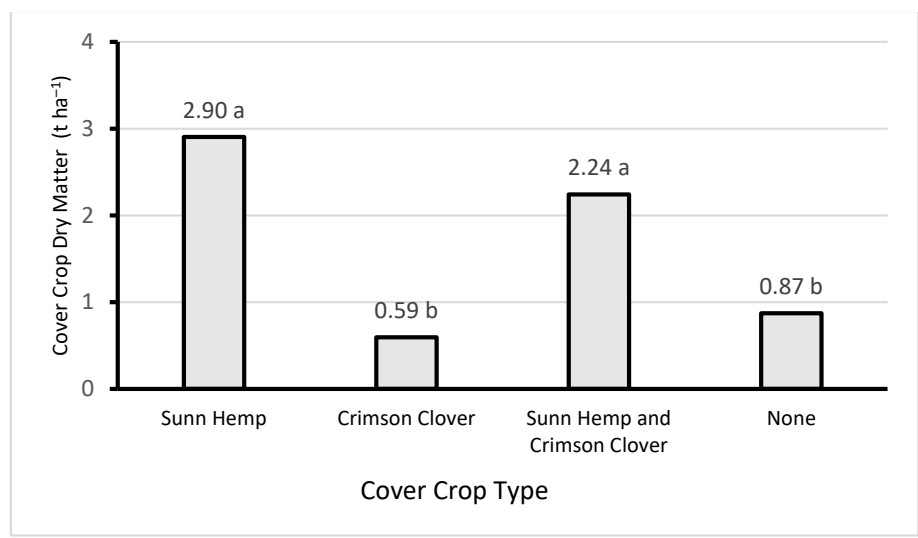

**Figure 1.** Mean aboveground cover crop biomass (t ha$^{-1}$) as a function of cover crop treatment. Means followed by the same letter are not significantly different from each other according to Bonferroni adjusted *t*-tests at $p \leq 0.05$.

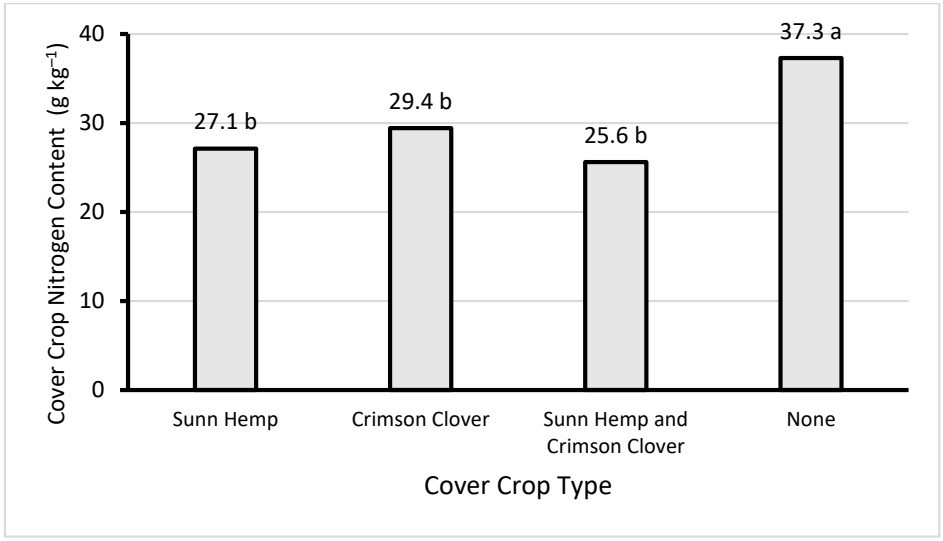

**Figure 2.** Cover crop biomass nitrogen content (g kg$^{-1}$) as a function of cover crop treatment. Means followed by the same letter are not significantly different from each other according to Bonferroni adjusted *t*-tests at $p \leq 0.05$.

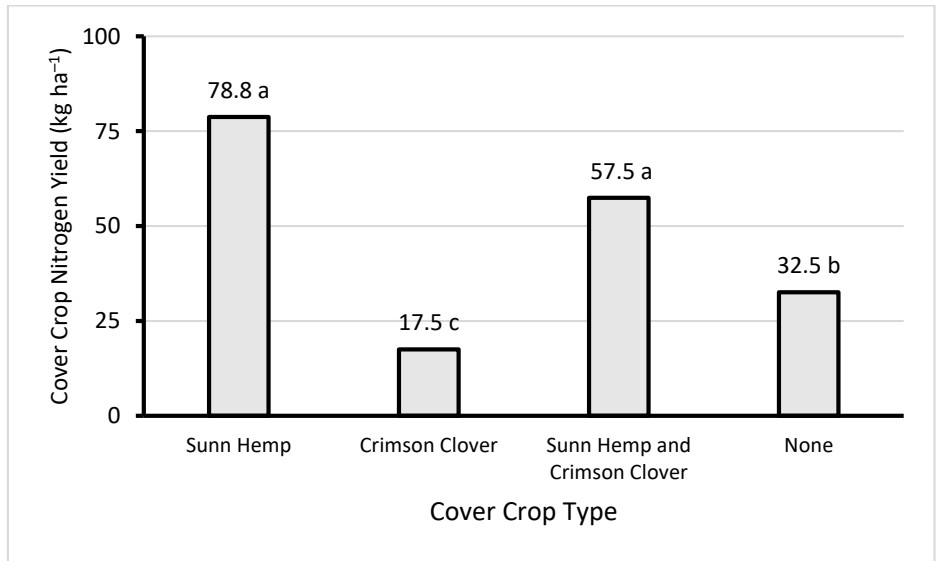

**Figure 3.** Mean aboveground cover crop nitrogen (kg ha$^{-1}$) as a function of cover crop treatment. Means followed by the same letter are not significantly different from each other according to Bonferroni adjusted *t*-tests at $p \leq 0.05$.

**Table 3.** Mean fall and spring soil nitrate following different cover crops in South Deerfield MA in 2014–2015 and 2015–2016.

| Cover Crop Type | Fall Soil Nitrate (mg kg$^{-1}$) | | Spring Soil Nitrate (mg kg$^{-1}$) |
|---|---|---|---|
| Sunn Hemp | 3.3 | a | 7.2 |
| Crimson Clover | 2.0 | b | 6.7 |
| Sunn Hemp and Crimson Clover | 2.8 | ab | 6.3 |
| No Cover Crop | 2.1 | b | 5.9 |
| Overall Experiment Mean | | | |
| | 2.5 | | 6.5 |
| Effect Significance | | | |
| Cover Crop Type | ** | | ns |

Note. **, $p \leq 0.01$; ns, non-significant according to non-parametric permutation tests. For significant effects, all pairwise comparisons were made using Bonferroni adjusted *t*-tests. Means followed by the same letter are not significantly different from each other ($p \leq 0.05$).

**Table 4.** Mean barley establishment, winter survival, and growth metrics at different barley seeding rates in South Deerfield MA in 2014–2015 and 2015–2016.

| Barley Seeding Rate | Barley Population (Plants m$^{-2}$) | | Winter Survival (Percent) | | Heading Date (Julian Day) | Height (cm) | Foliar Disease (Percent of Plants Affected) | Lodging |
|---|---|---|---|---|---|---|---|---|
| 300 seeds m$^{-2}$ | 288.5 | b | 95.3 | b | 139.9 | 54.1 | 52.5 | 10.4 |
| 350 seeds m$^{-2}$ | 355.0 | a | 97.8 | ab | 139.0 | 56.6 | 47.5 | 11.8 |
| 400 seeds m$^{-2}$ | 369.1 | a | 98.4 | a | 139.6 | 54.9 | 47.5 | 10.5 |
| Overall Experiment Mean | | | | | | | | |
| | 337.5 | | 97.2 | | 139.5 | 55.2 | 49.2 | 10.9 |
| Effect Significance | | | | | | | | |
| Barley Seeding Rate | *** | | ** | | ns | ns | ns | ns |

Note. **, $p \leq 0.01$; ***, $p \leq 0.001$; ns, non-significant according to non-parametric permutation tests. For significant effects, all pairwise comparisons were made using Bonferroni adjusted *t*-tests. Means followed by the same letter are not significantly different from each other ($p \leq 0.05$). Barley seeding rate was also evaluated as a.continuous effect using orthogonal polynomial regression.

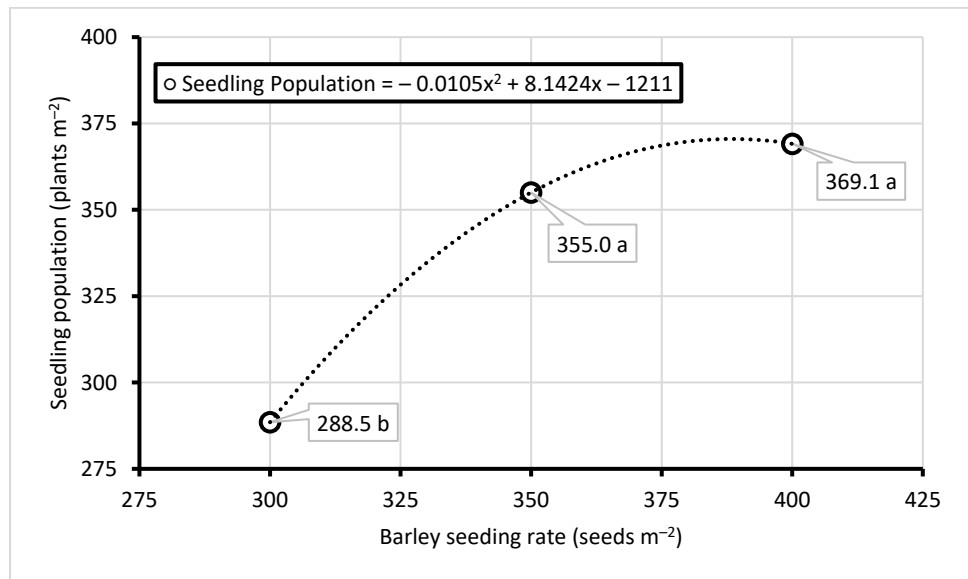

**Figure 4.** Mean seedling population (plants m$^{-1}$) as a function of barley seeding rate. Means followed by the same letter are not significantly different from each other according to Bonferroni adjusted *t*-tests at $p \leq 0.05$.

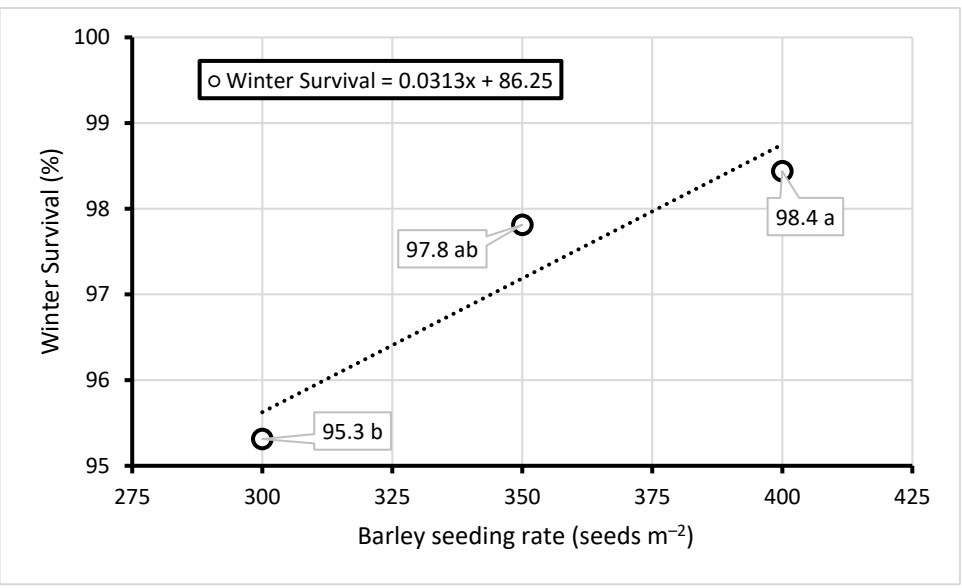

**Figure 5.** Mean winter survival (%) as a function of barley seeding rate. Means followed by the same letter are not significantly different from each other according to Bonferroni adjusted *t*-tests at $p \leq 0.05$.

Seeding rate did not have a significant impact on late vegetative and reproductive growth (Table 4). Seeding rate showed no influence on heading date or height of the barley plants, the severity of foliar disease, or the prevalence of crop lodging. All plots reached 50 percent heading within two days of each other and height averaged 55 cm across the experiment. Forty-nine percent of the leaves were affected by foliar disease and eleven percent of barley lodged before harvest across all treatments.

### 3.4. Influence of Cover Crop Species and Barley Seeding Rate on Barley Yield and Malting Quality Characteristics

There were no significant treatment effects on barley grain yield but there were two non-significant trends of note (Table 5). First, barley in SH and SH + CC treatments produced slightly more grain (3.74 and 3.67 t ha$^{-1}$, respectively) than either CC (3.39 t ha$^{-1}$) or NC (3.32 t ha$^{-1}$). Given that the cover crop treatment groups had very different organic matter and nitrogen contributions the previous fall, the small difference in yield could be related to the larger differences in soil condition at barley planting. Second, there was a non-significant yield improvement as seeding rate increased, possibly related to differences in stand quality due to differences in initial barley seedling population and winter survival. However, as noted above, the differences in yield were small and showed that all of the experimental treatments examined in this study can be similarly productive in malting barley cropping systems.

**Table 5.** Mean grain yield and quality metrics for winter malting barley following different cover crops and barley seeding rates in South Deerfield MA in 2014–2015 and 2015–2016.

| Cover Crop Type | Barley Seeding Rate (seeds m$^{-2}$) | Grain Yield (13.5% Moisture) (t ha$^{-1}$) | Protein (0% Moisture) (g kg$^{-1}$) | Test Weight (kg hl$^{-1}$) | 1000 Kernel Weight (g) | Germinative Energy (percent) | Falling Number (seconds) | DON (mg kg$^{-1}$) |
|---|---|---|---|---|---|---|---|---|
| Sunn Hemp | 300 | 3.53 | 118.4 | 58.68 | 46.40 | 80.38 | 233.9 | 0.20 |
| | 350 | 3.95 | 107.7 | 58.15 | 45.89 | 84.38 | 231.0 | 0.18 |
| | 400 | 3.75 | 110.0 | 58.06 | 45.55 | 79.75 | 234.4 | 0.10 |
| Crimson Clover | 300 | 3.15 | 114.8 | 58.66 | 47.00 | 85.00 | 237.3 | 0.17 |
| | 350 | 3.24 | 106.9 | 58.60 | 46.14 | 84.00 | 221.8 | 0.10 |
| | 400 | 3.77 | 114.3 | 57.90 | 46.70 | 80.00 | 227.0 | 0.20 |
| Sunn Hemp and Crimson Clover | 300 | 3.62 | 108.5 | 59.15 | 45.73 | 81.63 | 221.5 | 0.18 |
| | 350 | 3.85 | 111.0 | 58.28 | 47.13 | 89.13 | 212.1 | 0.15 |
| | 400 | 3.54 | 109.0 | 58.58 | 45.70 | 81.38 | 214.9 | 0.19 |
| No Cover Crop | 300 | 3.25 | 112.3 | 56.57 | 46.54 | 82.00 | 230.5 | 0.05 |
| | 350 | 3.23 | 111.1 | 58.85 | 46.43 | 79.38 | 211.5 | 0.15 |
| | 400 | 3.48 | 110.9 | 57.50 | 45.36 | 77.00 | 221.4 | 0.25 |
| **Cover Crop Type** | | | | | | | | |
| Sunn Hemp | | 3.74 | 112.0 | 58.30 | 45.95 | 81.50 | 233.1 | 0.16 |
| Crimson Clover | | 3.39 | 111.9 | 58.39 | 46.61 | 83.00 | 228.3 | 0.15 |
| Sunn Hemp and Crimson Clover | | 3.67 | 109.5 | 58.67 | 46.18 | 84.04 | 216.2 | 0.17 |
| No Cover Crop | | 3.32 | 111.5 | 57.64 | 46.11 | 79.46 | 221.1 | 0.15 |
| **Barley Seeding Rate** | | | | | | | | |
| 300 seeds m$^{-2}$ | | 3.39 | 113.5 | 58.26 | 46.42 | 82.25 | 230.6 | 0.15 |
| 350 seeds m$^{-2}$ | | 3.57 | 109.2 | 58.47 | 46.39 | 84.22 | 219.1 | 0.14 |
| 400 seeds m$^{-2}$ | | 3.64 | 111.1 | 58.01 | 45.83 | 79.53 | 224.4 | 0.18 |
| **Overall Experiment Mean** | | | | | | | | |
| | | 3.53 | 111.2 | 58.25 | 46.21 | 82.00 | 224.6 | 0.16 |
| **Effect Significance** | | | | | | | | |
| Cover Crop Type | | ns | ns | ns | ns | ns | * | ns |
| Barley Seeding Rate | | ns | ns | ns | ns | ns | ns | ns |
| Cover Crop Type × Seeding Rate | | ns | ns | ns | ns | ns | ns | ns |

Note. *, $p \leq 0.05$; ns, non-significant according to non-parametric permutation tests. Although cover crop type had a significant effect on falling number, pairwise comparisons made using Bonferroni adjusted *t*-tests did not show significant differences between any cover crop types ($p \leq 0.05$). Cover crop type is evaluated as a discrete effect and barley seeding rate as a continuous effect.

As was the case with grain yield, there were no significant differences in malting barley grain quality among cover crops or barley seeding rates (Table 5). Across the experiment, mean test weight was 58.2 kg hl$^{-1}$, mean 1000 kernel weight was 46.2 g, mean germinative energy was 82 percent, mean protein content was 111.2 g kg$^{-1}$, mean falling number was 224 s, and mean DON content was 0.16 mg kg$^{-1}$. Overall, while the barley met some malting quality standards, it fell below others. Protein and DON content were in good ranges for brewing purposes (below 125 g kg$^{-1}$ and 0.5 mg kg$^{-1}$, respectively). Test weight and falling number were a little lower than standard malting quality for these two indices (61.8 kg hl$^{-1}$ and 250 s) while germinative energy was much lower than required for

malting (95 percent). There is no specific standard for 1000 kernel weight. Relatively low germinative energy may have been the result of high drying temperatures following grain barley harvest.

These results indicate that high nitrogen contributions from leguminous cover crops are unlikely to result in protein levels exceeding malting standards.

## 4. Discussion

Most malting barley production in North America occurs in the dry Great Plains and West Coast regions with relatively little in the humid Northeast and Midwest [1]. As a result, most malting barley research has focused on different cropping systems than those discussed in this experiment. Many experiments have been performed in much drier conditions [16–20,22,24,39] or with spring planted cultivars [3,16,17,22,24,39]. Furthermore, many studies have found substantial differences in malting quality even in winter malting barley cultivars [10,11,14,18]. Caution should be used when generalizing experimental results given the diversity of growing conditions across North America and the relative novelty of winter malting barley cultivation in the Northeast.

This experiment indicated that even though malting barley is quite sensitive to nitrogen level in grains [13,14,18], leguminous crops may be grown before winter malting barley without damaging the yield or malting quality of the barley crop (Table 5). This result differs somewhat from research on malting barley production following legume cover crops further north in New England. Darby et al. and Surjawan et al. [3,23] reported that sunn hemp cover crop before winter barley reduced the next summer's yield in Vermont [23] and that a pea/oat/vetch cover crop slightly reduced malting quality in spring barley in Maine [3]. That said, malting barley is much more sensitive to nitrogen in the spring than in the fall [13] and Darby et al. [23] did not find effects of either sunn hemp or crimson clover on malting quality.

In the American West, past studies have explored whether growing spring malting barley after legume cash crops could impact the grain yield and malting quality [16,17,39]. Sainju [17] found that pea residue retained soil nitrate better than bare fallow, while Sainju et al. [39] and Turkington et al. [16] found that planting spring barley after peas did not cause negative quality characteristics such as high protein content, which have been commonly seen from excess nitrogen fertilizer [13,14,18]. However, these experiments were done with a spring barley following peas seed harvest, and the residues would not have contained nearly as much nitrogen as a legume cover crop incorporated into the soil, as was performed by Surjawan et al. [3].

The sunn hemp growth and nitrogen content (Table 2, Figures 1–3) in this experiment were similar to those seen after a similar amount of time (45 days) by Clark [25] in New York, although the overall sunn hemp production was lower than that seen by Clark after 60 days or by Mansoer et al. in the American Southeast [27]. Crimson clover (CC) and the summer fallow (NC) produced much lower biomass and nitrogen yield (Table 2, Figure 1; Figure 3) but did not lead to lower barley yields (Table 5). Together these results suggest that the barley was neither in need of nor hurt by the extra nitrogen supplied by the SH and SH + CC treatments. Given that malting barley can have disease issues when grown directly after a grass crop [1], it is agronomically important that winter malting barley be grown after a high nitrogen producer like sunn hemp without negative effects. Additionally, this research supports the idea recommended by Shrestha and Lindsey [1] that winter malting barley could be grown after soybeans (*Glycine max* (L.) Merr.). The growing season in Massachusetts may be too short for this to be a feasible cropping system. However, these results show that farmers further south should not be especially concerned that nitrogen from a previous soybean (or other legume) crop would negatively affect grain quality of winter malting barley. Additionally, longer growing seasons under climate change conditions and the development of shorter season soybeans could make this sort of crop system more attractive in the future in Massachusetts and other New England states.

While the differences seen in this trial based on barley seeding rate were minimal, previous studies have found that an increased seeding rate led to better overall malting quality and lower protein content in particular [20,22,24] in spring planted barley in western North America. The results from this experiment do not substantively contradict these earlier findings. While farmers are unlikely to see large yield gains from higher seeding rates, there may indeed be a reduction in variability from a relatively small investment in seed. Indeed, this research agrees with Darby et al. [23], who also found that similar increases in seeding rate can promote winter survival, although not always affecting final yield or malting quality overall.

The overall malting quality of barley measured in this study was similar to that seen in other winter malting barley in the Northeast. As in other studies, the barley grain did not meet all malting quality standards [11,13,23]. The best practices for malting barley production in the Northeast are still under development and producers should not expect to get malting quality grain every year [2,14]. Given that malting barley in the Northeast can be quite variable and is much more affected by cultivar selection [3,10,11,14], spring fertilizer application [13], and harvest management [2,12], the results of the current experiment indicated that regional farmers can plan their cropping system without worrying that a previous leguminous crop may cause quality issues in following winter malting barley.

## 5. Conclusions

It remains challenging to achieve superior malting barley in the Northeastern United States. This could lead farmers to form the impression that there is little flexibility if they want to meet the required quality standards. However, while excess nitrogen has been shown to lead to poor malting quality, growing high-nitrogen-producing legumes before planting winter malting barley is unlikely to reduce the quality of the succeeding barley. Specifically, this study demonstrates that following a legume cover crop with winter malting barley does not reduce grain yield or malting quality in terms of protein content, test weight, 1000 kernel weight, germinative energy, falling number, or DON content. These results show that winter malting barley can be integrated into crop rotations with leguminous plants without negative impacts on barley growth, yield, and grain quality.

**Author Contributions:** Conceptualization, M.H. and H.D.; supervision, M.H.; formal analysis, A.S. (Arthur Siller) and A.S. (Alexandra Smychkovich); writing—original draft preparation, A.S. (Arthur Siller); writing—review and editing, A.S. (Arthur Siller), M.H., H.D. and A.S. (Alexandra Smychkovich). All authors have read and agreed to the published version of the manuscript.

**Funding:** This study was partially funded by Northeast Sustainable Agriculture Research and Education, project number GNE 13-066.

**Data Availability Statement:** The data presented in this study are openly available at UMass Scholarworks.

**Conflicts of Interest:** The authors declare no conflict of interest. The funders had no role in the design of the study; in the collection, analyses, or interpretation of data; in the writing of the manuscript; or in the decision to publish the results.

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
