# Peer review of "Winter Malting Barley Growth, Yield, and Quality following Leguminous Cover Crops in the Northeast United States"

_nitrogen, doi:10.3390/nitrogen2040028_

Round 1
Reviewer 1 Report
Comments attached!

Author Response
Reviewer 1 remarks and responses
REVIEWER’S REPORT - Authors
Manuscript ID: Manuscript ID: nitrogen-1385471
Title: Winter malt barley growth, yield, and quality following leguminous cover crops in the Northwest United States.
General:
The study aimed at exploring the effect growing of leguminous cover crops will have on nitrogen uptake of winter barley and the quality of harvested malting winter barley. This will be a great idea if the study could lead to reduction of fertilizer application during the planting of winter barley.
However, it is important to note that many factors will affect the growing environment for cereals. The study is also important because interest is being shown in growing winter barley by plant breeders. From the abstract (line 10) and throughout the text, authors consistently used “malt barley”, I would suggest they use “malting barley”. In the abstract and in the conclusion, authors repeatedly stated that “there is much left to investigate in determining the best malting barley production......” I totally agree with this submission by the authors based on some of the specifics listed below.
response: 'malt barley' replaced with 'malting barley' throughout the manuscript, including in the title for Table 5.
Specifics:
Authors based their quality parameters on germination energy, test weight, and 1000-kernel weight. Although these parameters are useful and preliminary indicators of malting barley qualities, there are other more important parameters such as nitrogen content of barley and especially enzyme development during the malting of barley. There is an interesting paper “Utilisation of low-nitrogen barley for production of distilling-quality malt” published in the Journal of the American Society of Brewing Chemists in 2021 (Kirsty et al 2021). The paper will be of great importance for the authors to address the issue raised regarding the statement “there is much left to investigate in determining the best malting barley production......”. Most of the time, researchers tend to link high nitrogen content in malting barley to good enzyme development and good quality of malt. This is not always the case. It should be recalled that whilst all enzymes are proteins, not all proteins are enzymes. The paper, Kirsty et al 2021 addressed that issue.
Response: The authors thank the reviewer for bringing this article to our attention. The methods used to assess malting quality in this study were designed to follow malt house standards which crops would need to meet for sale as malting quality barley. It is exciting to see that new methods may refine the assessment of malting quality and the authors are highly interested in comparing these techniques to traditional methods in future studies.
Condition/s for accepting the manuscript for publication – My suggestion/s:
For the manuscript to be accepted, I will suggest the followings:
Since the authors agreed that “there is much left to investigate in determining the best malting barley production......”,
- The title should be modified, and I suggest “Winter malting barley growth, yield, and quality following leguminous cover crops in the Northwest United States – PART 1”.
- Authors should also clearly indicate that PART 2 of the study would be to carry out micromalting of the winter barley harvested from PART 1 study to determine the quality of malt produced from the barley – enzyme development, extract recovery and so on.
Response: Title changed to "Winter malting barley growth, yield, and quality following leguminous cover crops in the Northwest United States" as reviewer suggested.
Unfortunately, we only reserved enough barley grain from each plot to perform our planned analyses (plus some wiggle room) and would not be able to perform the additional malting quality analyses suggested as PART 2. As such, the authors do not feel that it would be correct to title this paper 'PART 1' since a follow up paper using micromalting techniques could not be done on the same barley samples. It would be highly useful to connect traditional malting quality metrics with micromalting methods. The authors would be interested in performing this type of research in the future but feel that it would have to be outside the scope of this project.
Reviewer 2 Report
The manuscript entitled Winter malt barley growth, yield, and quality following leguminous cover crops in the Northeast United States submitted by Siller et al. is well written. However, the reviewer has the following suggestions or queries; kindly incorporate and explain them before further processing of the article.
Remarks!
- The introduction part of the research article is well written but can be explained briefly without exaggerating it.
- DON stands for what? Please provide the full form where the abbreviation is used for the first time [Line 50].
- Kindly check line 75: [Darby 2016].
- Mineral nutrients’ amendments can be elaborated more [Lines 107-109].
- Please check lines 121-124.
- Barley stand was counted on 16 October 2015 but not in the fall of 2016 [Line 133]; what are the criteria for count Barley stand? Kindly explain.
- [Line 153-154] 2017 grain samples were stored in a cool seed room for several years before quality analysis. What does this sentence depict?
- Several results provided in the study were non-significant, is there any explanation for such observations.
- Kindly rewrite or recheck lines 294-296 and 302-306 to reflect the clear meaning.
- Please revise the conclusion part to summarize the outcomes of this work more appropriately.
Author Response
Manuscript ID: Manuscript ID: nitrogen-1385471
Reviewer 2 remarks and responses
The introduction part of the research article is well written but can be explained briefly without exaggerating it.
Response: The authors are trying to give a full picture of the status of winter malting barley in the Northeast USA. We feel that since barley is a relatively new crop in this region, some readers may not be familiar with malting barley production in general, and a full description of issues is useful. If the editors would prefer, the introduction could be abbreviated.
DON stands for what? Please provide the full form where the abbreviation is used for the first time [Line 50].
Response: Line 50-51, Line 160: definition added on line 50-51 and removed from line 164 so that it is defined at its first mention.
Kindly check line 75: [Darby 2016].
Response: Line 75: [Darby 2016] replaced with the correct reference formatting - [23]
Mineral nutrients’ amendments can be elaborated more [Lines 107-109].
Response: Line 108-109: specific mineral nutrient amendments added to text.
Please check lines 121-124.
Response: Line 124: sampling area corrected to 0.5 m2 from the incorrect typo 0.05.
Barley stand was counted on 16 October 2015 but not in the fall of 2016 [Line 133]; what are the criteria for count Barley stand? Kindly explain.
Response: Line 134-136: The data from 2016 was lost. There was not a decision made to skip stand counting in 2016. The text is changed to correctly reflect this.
[Line 153-154] 2017 grain samples were stored in a cool seed room for several years before quality analysis. What does this sentence depict?
Response: Line 156-158: clarification added to explain why germinative energy analysis was not included in mean values. Reference to the storage conditions before germinative energy testing is removed since it adds details which can be confusing.
Several results provided in the study were non-significant, is there any explanation for such observations.
Response: Overall, many parameters which we measured were not statistically significantly different between winter malting barley grown following legume cover crops and barley grown after a bare fallow. We believe that this is an agronomically significant result even though it is not statistically significant. Since barley grown after these summer legumes performs as well as barley grown after a fallow, there are additional crop rotation options for regional farmers. As this is a relatively novel crop regionally, many farmers have not had personal experience and research results are useful in showing that winter malt barley can be integrated into a variety of crop rotations without negative effects.
Kindly rewrite or recheck lines 294-296 and 302-306 to reflect the clear meaning.
Response: Lines 300-315: clarifications have been added to these lines to make the meaning clearer.
Please revise the conclusion part to summarize the outcomes of this work more appropriately.
Response: Lines 359-364: specific research outcomes added to the conclusion to clarify conclusion.
Round 2
Reviewer 2 Report
The authors have critically responded to my remarks. The current version of the manuscript can be accepted for publication.